# Linking Smart Governance to Future Generations: A Study on the Use of Local E-Government Service among Undergraduate Students in a Chinese Municipality

**Yonghan Zhu** *  and **Guitao Kou**

School of Public Affairs, Chongqing University, Chongqing 400044, China; Kouguitao@163.com
* Correspondence: zhuyonghan1992@126.com

**Abstract:** Due to the advanced technologies, governments today are facing more challenges in the governance field than ever before. One of the serious issues is how to develop relationships with younger generations. As a purpose of smart governance, sustainability emphasizes the responsibility of government for building a stable association with future generations. This study is devoted to promoting sustainability in the smart governance field through e-government services. It seeks to understand the situation of local e-government use in a special group of future generations: undergraduate students. In order to achieve this objective, this research conducts a case study in Chongqing, the only inland municipality in China. Drawing upon data from a sample of 1046 respondents in 2019, the findings reveal that the way to interact with the government via e-government is by receiving a wide range of undergraduate students at the local level. In this sense, the role of e-government in linking government and younger generations is larger and more significant than previously estimated. Additionally, the results witness a rise of social media in e-government services among younger generations. Based on all these findings, it offers practical suggestions for the future development of e-government services in China.

**Keywords:** sustainability in smart governance; e-government service; ICTs; e-government platforms; undergraduate students; social media

## 1. Introduction

Due to the development of information and communication technologies (ICTs), the way that the government interacts with citizens is sharply altering. Smart governance, as a rising governance model, provides a new mechanism for governments to build relationships with citizens [1–5]. Sustainability is one of most important objects of smart governance; it emphasizes the responsibility of government for taking into account the ecological implications of growth, improving the quality of life for future generations, and quickly responding to their citizens in cases of emergency [6].

Generally, smart governance is often related to e-government, another rising conception which refers to developing ICTs in public sectors in order to achieve administrative efficiency, service improvement, and citizen centricity [7–9]. The common belief is that e-government lays a great foundation for the development of smart governance [10]. Through those e-government services based on ICTs, the government can enhance the ability to promote sustainability and link with citizens more strongly [11].

Younger generations are commonly regarded as an important service target by the government. Improving the living qualities of young generation is indispensable step to achieve the sustainability of smart governance. With the development of ICTs, the Chinese government has begun a top–down

management reform in governance field. Those e-government platforms, such as websites and social media, have been participating in the delivery of local-service to all citizens, including undergraduate students [12]. Through those e-government platforms, local governments in China aim at strengthening their associations with older and younger generations to promote sustainability in the governance area.

Over last two decades, although many studies have focused on the adoption of e-government services among citizens [8,9,12], there are very few literatures evaluating the situation of e-government use among the younger generations, especially in China. As mentioned above, young people are important service targets of government. Thus, understanding the situation of e-government adoption among young people is significant for public sectors to improve their current e-government services and more effectively serve young citizens. This paper is devoted to understanding the development of e-government studies; it seeks to understand the situation of local e-government adoption among young people in China. This paper focuses on undergraduates, a specific group of the younger generation. In order to understand the situation of local e-government use among undergraduate students in China, this research conducted a case study in Chongqing, the only inland municipality of China where the local government has promptly reformed the delivery of local public service, but still confronted multidimensional inequalities rooted in a rural–urban divide [12].

The paper begins with a literature review to clarify those main concepts and their relationships in this study. Next, the data collection and measures are described. The results of this research find that currently, e-government is becoming a considerable tool to strengthen the link between government and younger generations. However, there is an unbalanced situation between local e-information use and local e-service use among undergraduate students, implying that the government needs to improve the current contents of e-government transactional services according to young people's main concerns. Additionally, it is suggested that public sectors in China should take into account more intelligent use of social media to achieve more in-depth political participation among undergraduate students, because this tool is more and more popular in this group of people. Finally, limitations and the conclusion are presented.

## 2. Literature Review

### 2.1. Smart Governance and E-Government

Currently, the role of the public sector in shaping society has shifted from government to governance, and the development of ICTs contributes to this process [13]. With the integration between ICTs and public administration, many new concepts are emerging. Smart governance is one of rising notions in this background; it is commonly understood as applying ICTs in the processing of information and decision-making in order to improve the capacity of governance. Core indicators of smart governance include data-based policymaking, featuring collaborative open and ICT-promoted transformation [14–25]. One of the main objects of smart governance is to achieve sustainability in the governance field, which means that public sectors should promote a long-term development to improve the quality of life for current and future generations [6]. In this sense, smart governance emphasizes a stable relationship between the government and citizens from generation to generation.

In general, e-government is defined as the promotion of ICTs in public sectors to improve administrative efficiency, quality of service, and citizen participation in decision-making processes [26–31]. Thus, e-government application supported by ICTs is devoted to increasing citizens' satisfaction, empowerment, and personal benefits, which in return strengthen the government–citizen relationships. Since those e-government services used to improve the efficiency of administration and service delivery offered to all citizens, including young groups, e-government services can also be devoted to strengthening the relationship between governments and future generations, and the stable relationship between governments and future generations can further promote the achievement of sustainability in the governance area. In this sense, the development of e-government lays the foundation for the development of smart governance [10].

Since e-government services can be regarded as an important tool to enhance the government–citizen relationship between the government and younger generations, a study on the situation of e-government use among young people can provide valuable data for the update of e-government services and the achievement of sustainability in the governance field.

### 2.2. Functions of E-Government Service

Basically, the major objective of e-government is to provide users with information and services [32–36]. In this sense, e-information and e-service are seen as the core functions of e-government service [37,38]. E-information refers to governments that provide policy-related, general, and service-related information for citizens through e-government tools [37]. E-service means that governments use e-government tools to offer transactional services and general services to citizens such as passport application, census registration, and certificate authentication [38].

Due to the development of ICTs, the types of e-government services are becoming varied. Recently, beyond e-information and e-service, the e-government has evolved to contain more functions, such as e-democracy and e-participation [37,39]. For instance, some research has pointed out that e-government services include three purposes: information provision, electronic transactions with government, and political participation [40,41]. Some other studies have focused on co-creation, a new function of e-government service that refers to the co-creation of policies, information, and services between governments and other citizens [42–46].

Although several decades of literature have found more and more types of e-government service, e-information and e-services are still seen as the most frequent and dominant functions [37,38,47]. Thus, the use of e-information and the use of e-services on the e-government platforms provides two important angles to understand the situation of e-government use among undergraduates in China. In this study, the situation of e-government use is composed of e-information use and e-service use. In addition, we conduct a comparison between the use of e-information and the use of e-services among undergraduate students in order to understand which major function of e-government is more popular.

### 2.3. User Satisfaction, User Frequency, Gender, and Age

Generally, user satisfaction and user frequency are regarded as the main factors to assess the situation of e-government use [12,48]. A high level of user satisfaction can be generated via good experiences of usage. In addition, when people enjoy e-government services, their usage behaviors tend to become more active and frequent [12]. Thus, user satisfaction and user frequency become most significant factors to evaluate the situation of e-government use among undergraduate students in this research.

To understand how citizens accept and react to e-government services, user acceptance models are made by e-government researchers according to information system theories. In 2003, the United Theory of Acceptance and Use of Technology (UTAUT) was developed by Venkateah et al. [49]. This theory argues that demographic factors, such as gender and age, have great impacts on the usage of individuals [12]. However, from the empirical studies testing the effects of demographic factors, evidences have proved opposite results: some found that men commonly have greater feelings of usage than women [50], while others pointed out that females scored higher than males in the usage of e-government services [51]. The same controversy also appears regarding age. Due to the uncertain effects of demographic factors, this paper also compares gender and age respectively in order to more clearly understand the situation of e-government use among undergraduates.

Based on the above arguments, we hypothesize that:

**Hypothesis 1a (H1a).** *The situation of e-information use is better than the situation of e-service use among undergraduate students.*

**Hypothesis 1b (H1b).** *The situation of e-information use is better than the situation of e-service use among male undergraduates.*

**Hypothesis 1c (H1c).** *The situation of e-information use is better than the situation of e-service use among female undergraduates.*

**Hypothesis 2 (H2).** *Compared with females, the situation of e-government use (including e-service and e-information) is better among males.*

**Hypothesis 3 (H3).** *Compared with undergraduates in lower grades, the situation of e-government use (including e-service and e-information) is better among those in higher grades.*

### 2.4. Local E-Government Service in China

In 2015, China started the "Internet Plus" strategy to promote socio-economic development. As an important section of this national strategy, "Internet Plus Government" was adopted in the governance field to provide citizen-centered public services through ICTs [12]. Since then, many local governments have published their e-government projects in the areas of public management, e-healthcare, citizen participation, etc.

Under the guidance of the "Internet Plus" Strategy, many e-government service platforms have emerged, and local e-government services in China have benefited greatly from these platforms. According to Porumbescu, e-government platforms can be divided into two types: a public sector website and social media [48]. In mainland China, "WeChat" from Tencent, "Alipay" from Alibaba, and "Weibo" from Sina have occupied social media markets. WeChat and Alipay are social media that contain comprehensive e-government services, such as messaging, e-payment, and e-information. Weibo is a micro-blog social networking application similar to Twitter [12]. According to a report by QuestMoblie, active users of WeChat and Weibo were around 800 million and 400 million respectively in 2016 [52].

This research is intended to understand the situation of local e-government use among undergraduate students in China. Therefore, e-government service (e-information and e-service) in this paper refers to local e-government service in Chongqing, China. Also, as one of the research objects, it seeks to find which type of platform is more significant to the e-government use among undergraduates. To address this question, WeChat, Weibo, and Alipay are chosen to constitute the type of social media. Official mobile apps in China are also included in this type, although it is not as popular as the former three platforms [53]. Thus, we hypothesize that:

**Hypothesis 4 (H4).** *As a type of e-government platform, the public sector website is as popular as social media among undergraduate students.*

## 3. Method

### 3.1. Data

The survey in this study was conducted in Chongqing municipality, China from April to May 2019. Chongqing was chosen as the target city because: (1) As the youngest municipality in China, Chongqing has promptly reformed the delivery of local public service via e-government. (2) These e-government projects can be more easily observed in a directly controlled municipality in China due to better autonomous ability and ICT infrastructure [12]. (3) Compared with the other three municipalities, Chongqing confronts significant multidimensional inequalities rooted in the rural–urban divide. Thus, Chongqing becomes an ideal case to understand the current situation of e-government service in regions with a rural–urban divide.

To improve the flexibility of the survey, paper questionnaires and online questionnaires were distributed. Paper questionnaires were distributed to undergraduate students in six local campuses and railway stations near the campuses. Online questionnaires were designed by an electronic survey software and were released through social networking software. The questionnaire started with demographic questions, such as sex, age, and grade. Then, respondents answered questions that measured the study variables, including the use of e-information and e-services, user satisfaction with e-government services, and user frequency of e-government services. The entire survey needs around 15 min to complete. In total, 1616 surveys were collected, including 312 paper surveys and 1304 online surveys. After discarding surveys that were incomplete, guaranteeing that the completed surveys matched the characteristics of the target sample based on occupations (undergraduate students), age, and gender, and discarding surveys that were finished in 5%< of the allocated time, totally 1046 surveys were valid: approximately a 64.7% response rate.

### 3.2. Measurements

#### 3.2.1. Experiences of E-Information

In this study, undergraduate students' e-government use is composed of their e-information use and their e-service use. Undergraduate students' e-information use is measured with two items: their experiences of e-information use and their satisfaction with e-information. The measurements to assess the experiences of e-information use are slightly modified from a previous study by Hong in 2013 [54]. In total, two items are employed to evaluate respondents' experiences of e-information use: the types of e-information services they used in the past year, and their user frequency of e-information services. The first item asks whether respondents have used the seven types of e-information services via e-government platforms in the past 12 months. Some of these services include "policies", "news about government", "news about other public sectors", "government circulars", "government forms", and the "way to contact public sectors" [37]. Each form of e-information service contains one score; a score of 0 indicates no use in the past year, and a score of 1 indicates that respondents have accessed this type of information through e-government platforms in the past year. The second item asks respondents to rate their frequency of e-information use in their daily life through a Likert scale that ranged from 1 (never use) to 5 (usually use).

#### 3.2.2. Experiences of E-Service

In this paper, respondents' situation of e-service use is measured with their experiences of e-service use and their satisfaction with e-service. In addition, the measurements to assess undergraduate students' experiences of e-service use were designed according to Hong's research [54]. A total of two items were used: the types of e-services they have used in the past 12 months, and their user frequency of e-service in daily life. The first item includes seven types of service as follows: "tax and social insurance payment", "passport application", "census registration", "certificate authentication", "drive license service", "payment of utility items (electronic power, water, gas, etc.)", and "complaint". Similarly, each type of e-service also contains one score; a score of 0 means no use in the past year, and a score of 1 indicates that respondents have experienced this service in the past year. Meanwhile, the user frequency of e-service is measured via a Likert scale that ranged from 1 (never use) to 5 (usually use).

#### 3.2.3. User Satisfaction

User satisfaction is another item to measure undergraduate students' situation of e-government use in this research. This item is also modified from Hong's study [54]. It evaluates the satisfaction through the question: "How satisfied do you feel with the e-information (e-service) you have used?" All the respondents' satisfactions with their use of e-government service were tested by a Likert scale that ranged from 1 (very dissatisfied) to 5 (very satisfied).

### 3.2.4. Website and Social Media

Website and social media are two types of e-government platform in this study. To understand which types of e-government platforms are more popular among undergraduate students, two items were employed. The first item asks students to choose the most usual platform they use to access information, and the second item asks students to select the most usual platform they use to experience government services. There are five options in each item: government website, Alipay, Wechat, Weibo, and Apps; respondents can only select one option in each item. Social media is composed of the latter four options. Through understanding which type of e-government platform is more popular in e-information and e-service, respectively, it can clearly conduct the comparison between websites and social media.

### 3.2.5. Age and Grade

Since the age differences are not significant among undergraduate students, grade becomes an important indicator to identify different groups of undergraduates. This study seeks to gather more data about the differences of e-government use between younger students and older students. Thus, all respondents were divided into four grades: freshmen, sophomores, junior students, and senior students, according to the common sense that older students are usually in higher grades. The bivariate correlations are shown in Appendix A.

## 4. Result

The demographic characteristics of respondents in the survey can be found in Table 1 as below. The ages of the respondents ranged from 17 to 25 years old, and 66.2% of them were between 18 and 21 years old. As for gender, 571 males and 475 females accounted for 54.6% and 45.4%, respectively. It also shows that students in liberal arts (66.6%) are overrepresented in the data. As for grade, the number of freshmen (228) is close to the number of sophomores (205), and the number of junior students (310) is nearly equal to the number of senior students (303).

**Table 1.** Sample characteristics ($N$ = 1046).

| Respondent Characteristics | $N$ | % |
|---|---|---|
| Gender | | |
| Male | 571 | 54.6 |
| Female | 475 | 45.4 |
| Age | | |
| 17 | 46 | 4.4 |
| 18–21 | 692 | 66.2 |
| 22–25 | 308 | 29.4 |
| Subject | | |
| Liberal arts | 697 | 66.6 |
| Science | 349 | 33.4 |
| Grade | | |
| Freshmen | 228 | 21.8 |
| Sophomores | 205 | 19.6 |
| Junior students | 310 | 29.6 |
| Senior students | 303 | 29.0 |

Table 2 reveals the differences between local e-information use and local e-service use among undergraduate students. According to Table 2, local e-information use is different to local e-service use among all respondents; undergraduate students seem to feel more enjoyable with e-information than e-services ($M_1$ = 9.4168, $M_2$ = 8.9321). Similarly, this tendency can be also found in both gender groups. Male students ($M_3$ = 9.4658, $M_4$ = 9.028) and female students ($M_5$ = 9.3579, $M_6$ = 8.8168)

show better feelings for the function of e-information than the function of e-service. These results offer support to Hypotheses 1a, 1b and 1c. In addition, Table 2 starts a consideration about the impact of gender differences on local e-government use. Compared with female students, local e-information use and local e-service use among male students show higher mean values, respectively. However, more analysis is still needed to understand the impact of gender differences.

Table 3 respectively examines how gender differences and grade differences influence the situation of local e-government use (including e-information and e-services). As it is shown in Table 3, although male respondents have a higher mean value of the situation of local e-government use ($M_7$ = 12.66) than female students ($M_8$ = 12.39), the F-value$_1$ is 0.670, implying that the impact of gender differences on the situation of local e-government use is not significant. Thus, there is no obvious differences between male students' situation and female students' situation; Hypothesis 2 is not fully supported. Additionally, the differences among four grades are not significant (F-value$_2$ = 1.469). It seems that sophomore students have the best situation of local e-government use, which is followed by junior students and senior students, while freshmen are ranked last. Therefore, no support is offered for Hypothesis 3.

**Table 2.** Paired sample test to examine the differences between the situation of e-information use and the situation of e-service use.

| | Results | Situation of E-Information Use | Situation of E-Service Use |
|---|---|---|---|
| Total | Mean | 9.4168 | 8.9321 |
| | Standard Deviation | 3.4936 | 3.7255 |
| | Paired Samples Test | t = 7.156 *** | |
| Male | Mean | 9.4658 | 9.0280 |
| | Standard Deviation | 3.5924 | 3.7710 |
| | Paired Samples Test | t = 4.932 *** | |
| Female | Mean | 9.3579 | 8.8168 |
| | Standard Deviation | 3.3738 | 3.6708 |
| | Paired Samples Test | t = 5.189 *** | |

Note: *** $p < 0.001$.

**Table 3.** One-way analysis of variance to examine the situation of e-government use with gender differences and grade differences.

| | Results | Gender | | Grade | | | |
|---|---|---|---|---|---|---|---|
| | | Male | Female | Freshman | Sophomore | Junior Student | Senior Student |
| Usage Situation | Mean | 12.66 | 12.39 | 12.18 | 13.06 | 12.71 | 12.26 |
| | F | 0.670 | | | 1.469 | | |
| | Significance | 0.413 | | | 0.221 | | |

In response to Hypothesis 4, more details about the use of local e-government service among undergraduate students were collected. Figure 1 reveals the relationship between e-government service and e-government platforms. It shows that 31.64% of respondents prefer to use the website to access information; this percentage is higher than any other social media platform, implying that the website is the most important channel to access information. However, if we saw Alipay, Wechat, Weibo, and official apps as a whole social media platform, the situation would be different. More than two-thirds of respondents (68.36%) select social media platforms to access information they need. As for e-service, social media platforms also account for more than two-thirds of the young market (68.93%). A point of note is that Wechat becomes the most common channel for undergraduate students to enjoy service. Although the use of social media is more and more common in the provision of local e-government

service, the website is still the most popular e-government platform to offer services (including 31.64% e-information and 31.07% e-service) among all the channels.

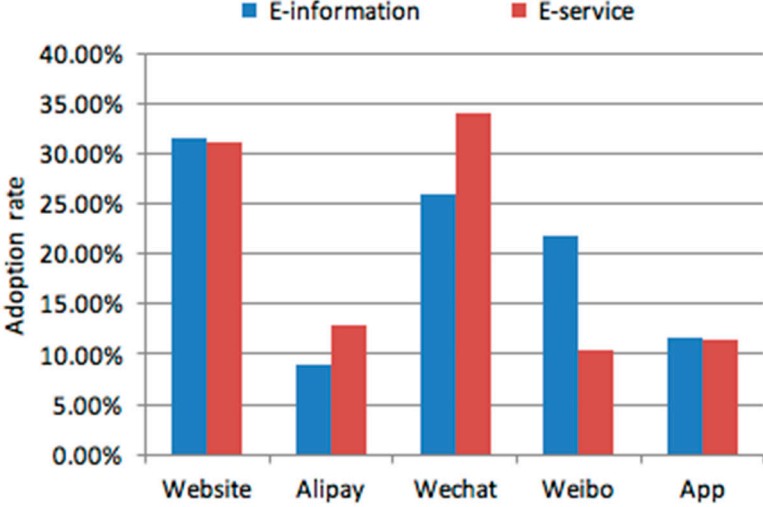

**Figure 1.** E-information in adoption of five channels and e-service in adoption of five channels.

Figures 2 and 3 respectively illustrate the relationship between gender and adoption rate and the relationship between grade and adoption rate. According to Figure 2, even though the adoption rates of websites are higher than any other channels in both genders, social media occupies nearly 70% of the young market. A similar situation can be found in grades; the adoption rates of website among freshmen, junior students, and senior students are less than 40% respectively, which means that the adoption rates of social media in the e-government service field is over 60%. In addition, it is worth mentioning that sophomore students only maintain around a 25% usage rate of the website. All the data from Figures 1–3 demonstrate that the website is less popular than social media in the e-government service area. Thus, Hypothesis 4 is not supported.

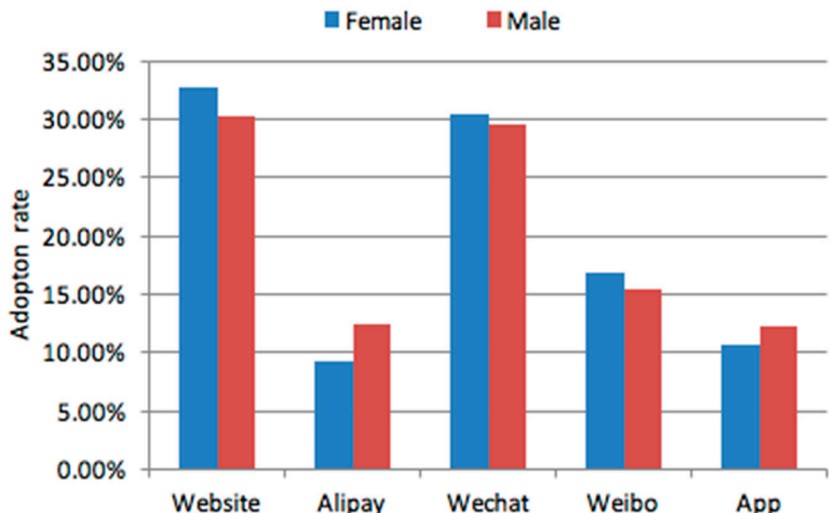

**Figure 2.** Gender differences in adoption of five channels.

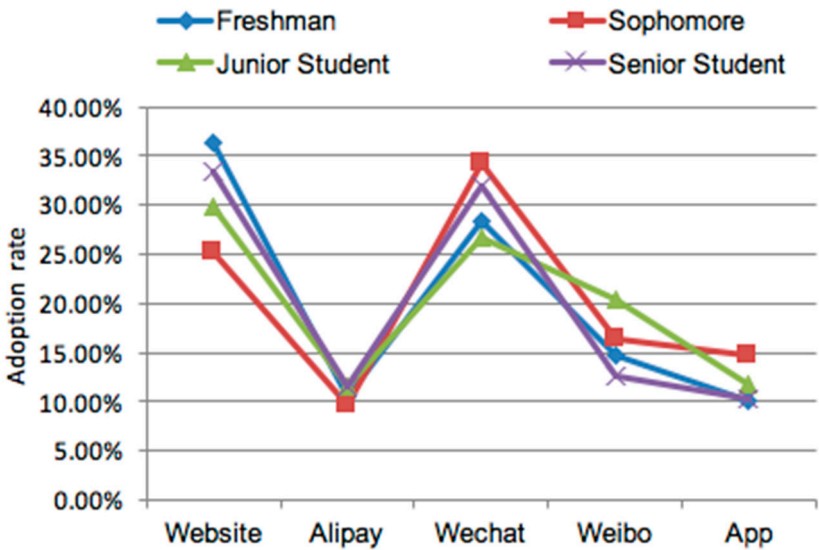

**Figure 3.** Grade differences in adoption of five channels.

## 5. Discussion

A great deal of literature has researched the utility of e-government service among citizens [54–56]. This study offers a specific perspective on undergraduate students, a group of the future generations that is regarded as important factor to promote the sustainability in the governance field. For an analysis, we proposed and empirically tested the situation of local e-government use among undergraduate students through the examination of their situations of e-information use as well as e-service use. In order to more clearly understand the situations, we examine the traditional factors of demography (age and gender) for predicting local e-government service adoption. In addition, the usage of e-government channels among undergraduates is tested in this study. Although there are many related studies about citizens' adoption of e-government service, it is useful to see one that focuses on undergraduate students. Thus, beyond the results that can be seen in much of the literature, this study also finds some unexpected results.

The e-inclusion model argues that there is a gap of user experiences between e-government information and e-government transactional services [57]. Thus, the situations of e-information use and e-service use among citizens are usually different. The findings in this study offer support to the e-inclusion model; the differences between the situations of e-information use and e-service use among undergraduates are significant. However, unlike some empirical work that pointed out that the situation of e-service use is better than the situation of e-information use [37], this study finds an opposite result among undergraduate students. The reason for this is probably because e-services in China fail to closely connect with undergraduate students' daily life. In contrast to e-information that can provide valuable information for undergraduate students, such as news and information related to employment, recreation, tourism, public policy, and government [37], those common contents of e-service, such as tax and social insurance payments, passport applications, census registration, payment of utility items (electronic power, water, gas, etc.), and complaints can be hardly used by Chinese undergraduate students. Therefore, compared with e-services, Chinese undergraduate students feel more enjoyment from the use of e-information. This result may ask local governments to update more of their e-service contents that are relevant to undergraduate students.

Many studies also have debated the impact of demographic factors on the usage of e-government services. This paper is devoted to offering more specific views on gender differences and age differences. First of all, the result demonstrates that there is no significant difference between males and females. This finding is similar to some in previous studies [58,59], but it is against other recent research that pointed out females make better use of digital channels than males aside from current inequalities in employment [60]. As for the impact of age, despite previous research holding the point that age is

positively associated with the usage of e-government services in the new digital era [12], the result shows that there are no obvious differences from freshmen to senior students. There are some alternative explanations for this finding. First, this result is possibly caused by insignificant age differences among undergraduate students. Since all of the respondents are between 17 and 25 years old, the narrow gap of their ages may have a slight impact on e-government use. The second explanation for this finding is probably that e-government services have already become a common tool for all undergraduate students in their lives. Since the adoption of e-government service has become a habit of students from all grades, their situation of e-government use tends to be similar to each other. In addition, the insignificant differences between males and females may also prove that e-government services have already been received by a wide range of undergraduate students in China. If so, e-government services would become an important bridge between the local governments and future generations in China.

Beyond these findings above, this paper also develops the research on the adoption of e-government platforms in China at the local level. Besides those Chinese platforms usually discussed in literature, such as websites, Wechat, Weibo, and official apps, this study includes another popular e-channel in China: Alipay. The findings are similar to some previous research on the usage situation of e-government platforms in China [12], indicating that social media is becoming the dominant platform for the interaction between e-government and citizens, even though websites still have a great impact. It is believed that this trend will be more clear with the growth of younger generations. Thus, the role of social media in promoting sustainability in the governance field is more and more significant. Questions such as how to effectively collect data via social media, how to achieve better decision-making through the analysis of data, and how to further improve the qualities of service and enhance the relation with the younger generation based on data have become great tasks for public sectors in China.

## 6. Conclusions and Limitations

Before moving onto discussing the conclusions, a discussion of the limitations in these measurements that pave the way for additional research is warranted. First, some of the demographic factors may not be covered in predicting the situation of e-government use among undergraduate students, such as subjects. Although many studies have demonstrated that educational backgrounds have a weak effect on e-government use [12,61–63], it is still worth evaluating this factor for more rigorous results. Thus, future study may consider educational backgrounds (education levels or subjects) when conducting the research on younger generations. Second, as mentioned above, the age differences are insignificant among respondents in this study. Actually, this problem will exist in most related studies on younger generations. Therefore, whether age differences are still needed to be accessed in such a research focuses on young people should be reconsidered. The shortages here could be remedied by future research that focuses on multidimensional studies of e-government use among younger generations. Third, the item of user frequency in this paper only assesses the whole situation of e-information (e-service) use and thus fails to accept the situation that a specific service may be normal to be used once or rarely. Therefore, it is important to investigate the frequency of every type of service in future research. In addition, the item of user satisfaction fails to investigate respondents' attitudes toward each type of service, and should be noticed by future study in this topic.

While this study is not without its limitations, our findings offer a deeper understanding on the usage of local e-government service among undergraduate students in China, and these findings can be applied to practical implications that promote sustainability in the governance field.

First, e-government service is becoming an important bridge to link local governments and younger generations. Nearly all of the undergraduate students in this study have used e-government services in their daily life. Second, it is now necessary for public sectors in China to focus on the design of the contents of e-service. Since more and more young people begin to receive e-government services, there is a great chance of reinforcing the relationship between local governments and younger generations through the services on e-government platforms. However, there is an unbalance situation

between e-local information use and local e-service use among undergraduate students, implying that the current contents of local e-government transactional services need redesign. In order to more effectively use e-government platforms to enhance the association with younger generations, the local governments in China should update local transactional services related to young people's main concerns. Through this way, the function of e-service can attract more young people and cultivate their trust in government. This is important to achieve the sustainability in the governance area.

Third, the findings highlight the role of social media in the provision of local e-government service in the future. Compared with websites, various social media channels together account for about two-thirds of the young market. These more convenient, user-friendly, efficient, creative, and diverse social media applications make local e-government services become easier and more accessible to undergraduate students, which in return attract a great deal of young users. One thing that should be noted is that males and females have nearly the same levels of adoption rate for every social media channel in this study. This indicates that these popular social media applications are well-designed to suit both genders. As mentioned above, the rising trend of social media in the e-government area cannot be neglected by government. Since there are more and more citizens using social media, these channels have great potential to collect numerous data for better decision making. Thus, it is important to update the current contents of e-service in social media and extend new functions based on the demands of young users. Through these approaches, younger generations will depend more on social media for local e-government services. On the other hand, it also should be noted that websites still have great impacts upon the usage of local e-government service among young people. This is possibly because a website has unique and irreplaceable characteristics as an e-government platform. For instance, a website is still the most formal channel to publish important announcements, policies, and documents. Therefore, public sectors should continue to emphasize the role of the website in e-government service delivery. Actually, with the development of e-government service in every aspect of our daily life, it is not difficult to predict a reintegration of functions of e-government service in the future. The website probably more focuses on the provision of political information, while general information and transactional services are more likely to be offered by social media.

Due to the advanced technologies, governments today are facing more challenges than ever before. To employ technologies to serve the public and achieve a sustainable relationship between government and citizens have become important purposes of smart governance. This study is devoted to these purposes by understanding the situation of local e-government use among undergraduate students in a Chinese municipality. As the largest developing country worldwide, China's situation may provide a great example for e-government research on younger generations. Thus, the findings in this paper can offer practical suggestions for the future development of e-government service.

**Author Contributions:** All authors discussed and agreed on the ideas and scientific contributions. Y.Z.: investigations, methodology, writing and editing; G.K.: investigations and data.

**Funding:** This research received no external funding.

**Acknowledgments:** This research could not have taken place without the great support in data collecting by the staff in the department of Public Administration, Chongqing University; the department of Chemistry, Chongqing University; the department of Biology, Chongqing University; the department of Automation, Chongqing University; the department of Information Technology, Chongqing University of Science and Technology; and the department of Design and Arts, Sichuan Fine Art Institute.

**Conflicts of Interest:** The authors declare no conflict of interest.

## Appendix A

**Table A1.** Bivariate correlations.

| #. | Variable | 1 | 2 | 3 | 4 | 5 | 6 | 7 |
|---|---|---|---|---|---|---|---|---|
| 1 | Gender | 1 | | | | | | |
| 2 | Grade | −0.049 | 1 | | | | | |
| 3 | Types of e-information | 0.012 | 0.055 | 1 | | | | |
| 4 | Types of e-service | 0.034 | −0.021 | 0.494 ** | 1 | | | |
| 5 | The situation of e-information use | 0.015 | 0.005 | 0.841 ** | 0.573 ** | 1 | | |
| 6 | The situation of e-service use | 0.028 | −0.038 | 0.531 ** | 0.870 ** | 0.818 ** | 1 | |
| 7 | Usage situation | 0.025 | −0.005 | 0.789 ** | 0.831 ** | 0.932 ** | 0.939 ** | 1 |

Note: ** $p < 0.05$.

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
