# Peer review of "Linking Smart Governance to Future Generations: A Study on the Use of Local E-Government Service among Undergraduate Students in a Chinese Municipality"

_informatics, doi:10.3390/informatics6040045_

Round 1
Reviewer 1 Report
The most valuable part of the paper is empirical research, however, it has a lot of limitations and the results can not be generalized to the young generation in China. The theoretical review is poor and have to be improved for defining more clear the gap in the literature and formulating the scientific question.
Also, the concept of e-governance lacks proper introduction. The main task of theoretical review is to provide knowledge about main concepts tested during the research and to build theoretical model for hypothesis formulation. The researcher has to prove that the hypothesis formulation is based on theoretical review, so the best way for it is to introduce each phenomenon (e-government, e-services, e-information, etc.), to explain, why and how each of the phenomenon is important to this research, to build theoretical model which clear links between phenomenons and only than to formulate hypothesis.
The hypothesis are formulated without interesting expectations, so the results are casual and do not offer interesting insights.
With many limitations the paper still can be published as a starting point for further research.
Author Response
Point 1: The researcher has to introduce e-government and explain why and how it is important to the research.
Response 1:The researcher has enhanced the introduction of the notion of e-government in following lines:
page 2 lines 18-38:
Currently, the role of the public sector in shaping society has shifted from government to governance, the development of ICTs contributes to this process [13]. With the integration between ICTs and public administration, many new concepts are emerged. Smart governance is one of rising notions in this background, it is commonly understood as to apply ICTs in processing of information and decision-making in order to improve the capacity of governance. Core indicators of smart governance include data-based policymaking, collaborative open and ICT-promoted transformation [14], [15], [16], [17], [18], [19], [20], [21], [22], [23], [24], [25]. One of main objects of smart governance is to achieve sustainability in governance field, which means public sectors should promote a long-term development to improve the quality of life for current and future generations [6]. In this sense, smart governance emphasizes a stable relationship between government and citizens from generation to generation.
In general, e-government is defined as the promotion of ICTs in public sectors to improve administrative efficiency, quality of service and citizen participation in the decision process [26], [27], [28], [29], [30], [31]. Thus, e-government application supported by ICTs devotes to increasing citizens’ satisfaction, empowerment and personal benefits, which in return strengthens the government-citizen relationships. Because those e-government services used to improve the efficiency of administration and service delivery are offered to all citizens, including young groups, e-government services can also devote to strengthening the relationship between government and future generation, and the stable relationship between government and future generation can further promote the achievement of sustainability in governance area. In this sense, the development of e-government lays the foundation for the development of smart governance [10].
Point 2: The researcher has to explain e-services and e-information and explain why they are important to this research.
Response 2: The researcher has enhanced the explanation in following lines:
Page 2 lines 43-49; Page 3 lines 1-15:
Basically, the major objective of e-government is to provide users with information and services [32], [33], [34], [35], [36]. In this sense, e-information and e-service are seen as the core functions of e-government service [37], [38]. E-information refers to governments provide policy-related, general and service-related information for citizens through e-government tools [37]. E-service means governments use e-government tools to offer transactional services and general services, such as passport application, census register and certificate authentication, to citizens [38].
Due to the development of ICTs, the types of e-government services are becoming various. Recently, beyond e-information and e-service, e-government has evolved to contain more functions, such as e-democracy and e-participation [37], [39]. For instance, some research has pointed out e-government services include three purposes: information provision, electronic transactions with government and political participation [40], [41]. Some other studies have focused on co-creation, a new function of e-government service which refers to co-creation of policies, information and services with government and other citizens [42], [43], [44], [45], [46].
Although several decades’ literatures have found more and more types of e-government service, e-information and e-service are still seen as the most frequent and dominant functions [37], [38], [47]. Thus, the use of e-information and the use of e-services on the e-government platforms provides two important angles to understand the situation of e-government use among undergraduates in China. In this study, the situation of e-government use is composed of e-information use and e-service use. Also, we conduct a comparison between the use of e-information and the use of e-services among undergraduate students in order to understand which major function of e-government is more popular.
Reviewer 2 Report
3.1. Data
Line 27 :What were the criteria a survey to be valid?
Line 38: Why 1 year was set as a time period? How often this information is being updated?
3.2.2. Types of Services. The undergraduate students were adults? Were they obliged-allowed to use the services e.g. tax, drivers license etc. Is it clear that they used the service in general in the last year e.g. with physical presence? If they did not plan to use it then how can we conclude something about the e-service use?
3.2.3. User satisfaction and user frequency
Is user frequency applicable in all services e.g. drivers license. Was there any discrimination regarding this aspect?
3.2.4. Website and social media
Line 14, 15 Why select only once option. A combination is not applicable?
Result
Page 6: What is the level of confidence of the results presented? Was the sample representative? Can these results be generalized?
General remarks
It would be interesting to understand in the specific sample, how the undergraduate students react in e-services in general and if the use of e-government services in higher or lower comparatively to the general use of e-services.
It would be also interesting to understand the environment that the undergraduate students belong to. E.g. do they have internet access in their home, Do they have access only in the University. Are they working in parallel with their studies etc. This will help us to understand their level of involvement with the government services and the facilities that they can exploit for the use of e-government services.
Author Response
Dear reviewer:
Thanks very much for your suggestions!
Data
Point 1: What were the criteria a survey to be valid?
Response 1: Thanks very much for the suggestions. The research develops three criteria to assess the valid survey.
We discard surveys that were incomplete We make sure that the completed surveys matched the characteristics of target sample based on occupations (undergraduate students), age (those who over 18 and those who currently are 17 but will soon be 18 in 2019) and gender. We discard surveys that were finished in 5%< of the allocated time.
We have written these criteria in the paper.
Point 2: Why 1 year was set as a time period? How often this information is being updated?
Response 2: The purpose of this paper is to understand the current situation of e-government use among undergraduate students. We do not plan to trace a process of change among undergraduate students in 2 or more years, so we only set 1 year as a time period. These results in the paper will become important data for future studies in this topic. Additionally, the information in this study was updated every day during the period of collection.
Point 3: The undergraduate students were adults? Were they obliged-allowed to use the services e.g. tax, drivers license etc. Is it clear that they used the service in general in the last year e.g. with physical presence? If they did not plan to use it then how can we conclude something about the e-service use?
Response 3: All the respondents in this paper are adults. Actually, there are 46 respondents with age 17 in the paper. All these data (46 respondents) were collected from paper survey and we understood that these 46 respondents are 17 now but will soon be 18 in 2019. If the respondents did not plan to use the services, we conducted the "frequency of use" to assess them. Respondents were asked to rate their frequency of e-government use through a Likert scale that ranged from 1 (never use) to 5 (usually use). Those who never use a service were asked to select 1 score.
Point 4: Is user frequency applicable in all services e.g. drivers license. Was there any discrimination regarding this aspect?
Response 4: For those who never use such services like driver license, they were asked to choose 1 score in user frequency item. Also, in this paper, online drivers license service include the whole process of getting a driver license: the register for tests (there are totally four tests in China, the candidates must pass one test and then can enter the next one. Also, the candidates can refuse to use the online register services, they can ask their training agencies to help register), the online learning services (free, including videos and other methods, people can use the services by their own), etc. Thus, user frequency is applicable in the driver license services, like "how often do they use the online learning services" and "how often do they choose the online register system".
Point 5: Why select only once option. A combination is not applicable? (Social media and website)
Response 5: The "social media and website" are based on Porumbescu's previous paper 'Linking public sector social media and government website use to trust in government'. (2016). In the paper, Porumberscu used two platforms, social media and government website to assess the use of e-government. Additionally, the combination between social media and government website in current China's e-government services is not applicable. Many services in social media cannot be used in government website, and also some policies can only be found in website. Thus, this paper argues that there perhaps is a reintegration of functions of e-government service in the future. Website probably more focuses on the provision of political information, while general information and transactional services are more likely to be offered by social media.
Point 6: Page 6: What is the level of confidence of the results presented? Was the sample representative? Can these results be generalized?
Response 6: The data of this research were collected from one of national cities in China and 7 departments have taken part in the data collecting. The respondents come from different parts of China. Thus, it is believed that the results have values.
Round 2
Reviewer 2 Report
What were the criteria a survey to be valid? O.K.
Line 40: repetition “discarding”
Why 1 year was set as a time period? How often this information is being updated?
The Digital Society Index of EU get the view from the regular Internet users, which has value.
One year time period includes users that are not regular and therefore they do not have a clear view about the services. According to my perception their responses cannot be considered as reliable.
The response of the authors did not provide any explanation for that. The frequency Likert scale is not clear if it is used as an index for the reliability of the response. It seems to be used only as an information.
3.2.2. Types of Services. The undergraduate students were adults? Were they obliged-allowed to use the services e.g. tax, drivers license etc. Is it clear that they used the service in general in the last year e.g. with physical presence? If they did not plan to use it then how can we conclude something about the e-service use?
3.2.3. User satisfaction and user frequency
Is user frequency applicable in all services e.g. drivers license. Was there any discrimination regarding this aspect?
According to the authors response the user Frequency was applicable in every service without accepting the situation that a service may be normal to be used once or generally rarely and therefore the service should be treated differently in this type of question. It seems that there was not discrimination for that,
3.2.4. Website and social media
Line 14, 15 Why select only once option. A combination is not applicable?
O.K.
Result
Page 6: What is the level of confidence of the results presented? Was the sample representative? Can these results be generalized?
It seems that the sample was random and no statistical criteria were applied. My impression is that if we repeat the same survey in the same area with the same approach, we will get different – random results. Therefore, the level of confidence seems to be very low. I do not believe that these results can be generalized.
Author Response
Dear reviewer,
Thanks very much for your suggestions! We found your comments most helpful and have revised the manuscript. We have sent the revised manuscript according to the comments of you. Revised portion are underlined in red. Also, please allow us to response some of your points:
Point 1: Why 1 year was set as a time period?
Response: This is very great point. First, please allow us to apologize for the English writing. In the original paper, we wrote “use e-information (e-service) within a year.” Actually, this is not what we mean, the real question we asked was “use e-information (e-service) in the past year (12 month).” We do not employ “one year” to assess the situation of adoption, but we want to assess the recent situation of e-government use (one year).
Please allow me to explain more about this. The measurements of this paper are based on the pervious study “Government websites and social media’s influence on government-public relations” by Hong in 2013. The article is published on “Public Relations Review”. In the article, Hong employed two item to respectively assess respondents’ experiences of e-information (e-service) use: the types of services (e-information and e-service) they have used during the past 12 month.
“Experiences with government websites were further explored, and for this, some questions regarding their use of information services and transactional services were selected. Specifically, respondents were asked a set of yes/no questions about the use of 15 types of information services through government websites during the past 12 months …… The second type of government website was transactional service, which was measured using three questions. Respondents were asked whether in the past 12 month, they had (1) renewed his or her driver’s license or registration online……” (Hong, 2013, pp. ).
Thus, we also use the same way to assess the usage of types of e-government and we investigate the situation “in the past year”. In the revised manuscript, we have modified “within one year” to “in the past year”.
Point 2: About user frequency
Response 2: This is great point which finds the major problem in measurement. First, we made major modification in measurement. We used two items to assess the situation of e-government use: the experiences of e-government use and the satisfaction with e-government services (the original paper includes three items).
The experience of e-government use is slightly modified from Hong’s study. In Hong’s paper, the experience of e-government use only contains the usage of types of e-government services. In our paper, we use another item to evaluate the experiences: user frequency. However, the user frequency refers to the user frequency of all information or transactional services. This item fails to evaluate the frequency of every services and thus neglect the situation that some respondents may only use some services once or rarely, as you pointed out. So, we highlight this in the limitation part to pave the way for future studies (We also highlight the problem of satisfaction in the limitation part).
If you considered that the user frequency was still not appropriate, we could delate this item and only use the types of e-government use as Hong’s did.
Point 3: About the results
Response: As you pointed out, the survey was conducted with the same approach. But we think our original title makes you think the confidence is low. Actually, our paper is “a case study in a Chinese Municipality”. We also do not think the results can represent the whole situation in China, but we do agree that the results can represent the real situation in Chongqing, the only inland municipality in China, because we conducted the survey in nearly every major Universities of Chongqing with the official support in these Universities. Actually, our research focuses on the local development of e-government in China, as the 2.4 in literature review part has represented: “local e-government service in China”. Our questions about information and transactional services only concern local e-government services. Thus, we modified the title of the paper, we think our title should more focus on the development of local e-government. We change the title to “Linking smart governance to future generation. A study on the use of local e-government service among undergraduate students in a Chinese Municipality”. We also notice that our questions only concern local e-government services in the paper.
Some of previous studies about the e-government use, such as Yang’s paper “Towards a New Digital Era: Observing Local E-government Services Adoption in a Chinese Municipality” (published on “Future Internet”), also used the same way to collect data. In Yang’s research, both online questionnaire and offline questionnaires were distributed to citizens randomly in railway stations and main streets with crowded population in Chongqing. Also, In Karunasena’s and Deng’s article “Critical factors for evaluating the public value of e-government in Sri Lanka” (published on “Government Information Quarterly”), they conducted paper-based survey in Sri Lanka and distributed around 1000 papers randomly. Thus, we think the confidence of our results is not very low, the results can represent the real situation about the e-government use among young people at local level.
As you have pointed out, there are many limitations in this paper, but we still believe it can devote to future studies on this topic. Currently, there are very few studies on the e-government use among young generation.
Please see the attachments. Thanks for your comments again!
Kind regards,
Yonghan

Round 3
Reviewer 2 Report
Please see page 10 line 11 assesses instead of assess
Author Response
Dear reviewer,
Thanks very much for your comments. We are sending the revised manuscript according to your comments (We have modified the "assess"). Revised portion are underlined in red.
Thanks again!
Kind regard,
Yonghan
